# Carpal tunnel syndrome and occupational co-exposure to biomechanical factors and neurotoxic chemicals using job-exposure matrices and self-reported exposure: Findings from the Constances cohort

Julie Bodin[1]*, Clémence Rapicault[1], Alexis Descatha[1,2], Marc Fadel[1], Fabien Gilbert[3], Bradley Evanoff[4], Nathalie Bonvallot[5], Marcel Goldberg[3], Marie Zins[3], Yves Roquelaure[1]

1 Univ Angers, CHU Angers, Univ Rennes, Inserm, EHESP, Irset (Institut de recherche en santé, environnement et travail), Angers, France, 2 Department of Occupational Medicine, Epidemiology and Prevention, Donald and Barbara Zucker School of Medicine, Hosftra University Northwell Health, New York, United States of America, 3 Paris Cité Université, "Population-based Cohorts Unit", INSERM, Paris Saclay University, UVSQ, Paris, France, 4 School of Medicine, Division of General Medicine and Geriatrics, Washington University in St. Louis, St. Louis, Missouri, United States of America, 5 Univ Rennes, Inserm, EHESP, Irset (Institut de recherche en santé, environnement et travail), Rennes, France

* Julie.bodin@univ-angers.fr

## Abstract

### Objective

To study the association between occupational co-exposure to biomechanical risk factors, potentially neurotoxic chemicals and carpal tunnel syndrome (CTS) in a large cohort of French workers, using two methods to estimate chemical exposure: job-exposure matrices (JEM) and self-reported exposure.

### Methods

A randomly selected sample of adults were included between 2012 and 2018 in the French cohort CONSTANCES. Self-reported CTS was assessed using the first self-administered follow-up questionnaire, sent out approximately one year after baseline. Occupational exposure to biomechanical risk factors was assessed using self-administered questionnaire completed at inclusion. Lifetime occupational exposure to chemicals was assessed using two different methods: with JEMs and with a self-administered questionnaire completed at inclusion. Multivariate logistic regression models were used to evaluate the association between co-exposure to biomechanical risk factors and chemicals and CTS, adjusted for personal and medical factors and stratified by gender.

**Data availability statement:** The data underlying this article were provided by the CONSTANCES cohort under permission and are not publicly available, to ensure the confidentiality of study participants. However, the CONSTANCES cohort is "an open epidemiological laboratory" and access to study protocols and data is available on request (https://www.constances.fr/en/scientific-area/access-to-constances-2/)

**Funding:** The Constances cohort study was supported and funded by the French national health insurance fund ("Caisse nationale d'assurance maladie", Cnam). Constances is a National infrastructure for biology and health ("Infrastructure nationale en biologie et santé") and benefits from a grant from the French national agency for research (ANR-11-INBS-0002). Constances is also partly funded to a small extent by industrial companies, notably in the healthcare sector, within the framework of Public-Private Partnerships (PPP). This work was also supported by the French National Research Program for Environmental and Occupational Health of the Anses, Maison-Alfort, France (ANSES-21-EST-030). None of these funding sources had any role in the design of the study, collection and analysis of data or decision to publish.

**Competing interests:** The authors have declared that no competing interests exist.

## Results

For the analysis using JEM assessment, 35,941 workers (16,920 men and 19,021 women) were included: 261 men (1.5%) and 469 women (2.5%) declared having CTS at follow-up. There was an association between CTS and the co-exposure group: OR=2.37 [1.60–3.44] in men and OR=2.09 [1.55–2.77] in women, compared to the non-exposed group. For the self-reported chemicals analysis, 42,168 workers (20,877 men and 21,291 women) were included: 338 men (1.6%) and 532 women (2.5%) declared having CTS at follow-up. There was an association between CTS and the co-exposure group: OR=3.07 [2.28–4.08] in men and OR=2.68 [1.91–3.66] in women, compared to the non-exposed group.

## Conclusions

The study showed an association between self- reported CTS and co-exposure to biomechanical risk factors and chemicals. This finding should be confirmed using more objective case definition of CTS, e.g. carpal tunnel release surgery.

## Introduction

Carpal tunnel syndrome (CTS) is a common entrapment neuropathy in the general population and in the working population [1], with repercussions on personal and professional life. Several studies have highlighted associations between CTS and personal characteristics (e.g., advanced age, female gender) [2,3], medical conditions (e.g., obesity, diabetes mellitus, arthritis) [4–6] and occupational exposure to biomechanical factors (e.g., repetitive movements, hand-arm transmitted vibration, and forceful manual exertion) [1,7–13]. There is limited epidemiological evidence for an association between psychosocial factors (occupational or not) and CTS [14].

Multi-exposure situations are common in the world of work, whether they involve exposure to multiple biomechanical or chemical factors, or both at the same time. In previous studies, we showed that occupational co-exposure to biomechanical factors and neurotoxic chemicals was observed among 4.8% to 10.9% of men and 0.7% to 3.7% of women [15,16]. Workers under 30 years old, blue-collar workers and those working in small companies are more often co-exposed to biomechanical stressors and chemical agents [16]. Some of the chemical agents encountered in the workplace have neurotoxic properties that can alter the functioning of the central nervous system (CNS) and/or the peripheral nervous system (PNS) [17–25]. To the best of our knowledge, the impact of chemical exposure on the risk of CTS has rarely been studied despite the potential neurotoxicity of some chemicals [15,26–31]. Ophir et al. [29] reported an increased risk of CTS-like symptoms following subclinical sensory polyneuropathy (affecting mainly the median and sural nerves) in workers exposed to prolonged low-level exposure to organophosphate in rural communities in Israel. However, a case-control study of CTS conducted in the general population of Wisconsin failed to report an association between CTS and chemical exposure after

adjustment for the known risk factors [26]. The frequent biomechanical and chemical co-exposure raises the question of potential synergistic effects of mechanical stressors and neurotoxic chemicals on the risk of CTS [15]. Systemic peripheral neuropathies (e.g., diabetic polyneuropathy) are known to cause subtle nerve damage, thus rendering the median nerve more vulnerable to compression within the carpal tunnel and against the volar ligament in the case of overexposure to biomechanical stressors [32,33]. Following the same reasoning, exposure to chemicals in the workplace could also cause subclinical changes in the median nerve [29] – producing a form of double crush syndrome that could weaken the median nerve [34]. This could potentiate mechanical stress on the nerve in the event of co-exposure with biomechanical stress factors that increase intra-tunnel pressure. Subclinical changes in the central nervous system (CNS) can also occur in workers exposed to chemicals [35,36] and lead to impaired sensorimotor control of force production and finger dexterity, thus generating greater mechanical stress on the median nerve.

In previous studies, we have shown an association between CTS and co-exposure to biomechanical risk factors and chemicals, after adjustment for confounders in a cohort of French workers [15] and in a cross-sectional study of French Farmers and Agricultural Workers [31]. Lifetime exposure to chemicals has been assessed retrospectively, which may lead to misclassification of exposure. The present study aimed to investigate the association between co-exposure to bio-mechanical risk factors and potentially neurotoxic chemicals and incident CTS in a large cohort of French workers, using two methods to estimate chemical exposure: job-exposure matrices (JEM) and self-reported exposure.

## Methods

### Population

The Constances cohort (doi.org/10.13143/inserm_constances) is a French epidemiological cohort of randomly recruited participants aged 18–69 years at baseline [37]. Eligible participants were selected from residents of the departments where the 21 Social Security Health Examination Centers (HSC) involved in the study are located. At inclusion, between 2012 and 2018, the subjects are invited to complete questionnaires (Lifestyle and health, Women's health, Job history and Occupational exposures) and to attend a HSC for a comprehensive health examination. During the follow-up, an annual questionnaire is sent to the participants, in paper form or online, according to the volunteers' choice. Data extracted by the Constances team in 30/11/2021 were analyzed.

Two analyses were conducted: one using JEM and another using self-reported exposure to estimate lifetime occupational chemical exposure.

For both analyses, we selected participants who responded to the first follow-up questionnaire between six and eighteen months after inclusion in the cohort. We excluded: 1) non respondents to Lifestyle and health and Occupational exposures questionnaires at baseline, 2) self- reported CTS or chronic hand symptoms at baseline, 3) participants professionally non active at baseline and/or follow-up, 4) pregnant women at baseline and/or follow-up, and 4) workers aged 65 years or over (S1 Fig). In addition, for the first analysis (JEM), we excluded workers whose job history could not be linked with the JEMs (due to incomplete job codes or industry sector codes) and workers with missing data for occupational exposure (biomechanical and chemical). For the second analysis (self-reported), workers with missing data for occupational exposure (biomechanical and chemical) were excluded (S1 Fig).

All volunteers signed a written consent form for their participation in CONSTANCES. The CONSTANCES cohort was authorized by the French personal data privacy authority (CNIL #910486) and was approved by the Institutional Review Board of INSERM (IRB INSERM #01–011 and 21–842).

### Outcome at follow-up

At the 12-month follow-up, incident CTS was assessed by answering the following question: "*Do you suffer or have you suffered from CTS in the last 12 months (whether CTS required sick leave or not and/or treatment or no*t)?".

## Biomechanical exposure at baseline

Biomechanical exposure relevant to CTS was defined as exposure to at least one of the five following factors in the 12 month period preceding baseline (yes/no) [1,7–12]: high perceived exertion (>12 on the Borg rating of Perceived Exertion scale, graduated from 6-'very, very light' to 20-'maximum exertion'), repetitive hand movements (more than 4 hours/day), hand-transmitted vibrations (more than 2 hours/day), awkward wrist postures (more than 2 hours/day), holding tools/objects in a pinch grip (more than 4 hours/day). The thresholds were defined according to the European criteria document for the relatedness of MSDs [38] and a INRS (French National Research and Safety Institute for the Prevention of Occupational accidents and Diseases) report [39].

## Lifetime chemical exposure

Chemicals studied were selected according to their potential neurotoxicity [17,40–44] and their availability in JEMs [45] and self-administered questionnaires. At the time the analyses were carried out, only the JEMs for chlorinated solvents and formaldehyde were available in the same nomenclatures as the Constances job codes. Chlorinated solvents are known as neurotoxicants [22,44,46–48]. The neurotoxicity of formaldehyde may involve several pathways, including inflammation or overproduction of intracellular reactive oxygen species (ROS) [40,41].

JEMs were developed for the French general population in the context of the Matgéné program by Santé publique France [45]. The chlorinated solvents JEM assesses the probability of exposure to at least one of the five chlorinated solvents (perchloroethylene, trichloroethylene, methylene chloride, chloroform, and carbon tetrachloride) from 1950 to 2021 for combinations of activity sectors (Nomenclature d'Activité Française, NAF) and occupations (Professions et Catégories Socioprofessionnelles, PCS) codes. The formaldehyde JEM assesses the probability of occupational exposure from 1950 to 2018 for combinations of NAF and PCS codes. The JEMs were linked with lifetime occupational history, and lifetime chlorinated solvents and formaldehyde exposures were assessed as follow: never exposed and ever exposed (defined as having worked in at least one job – combination of PCS and NAF codes – with a probability of exposure greater than zero). For exposure to at least one solvent, cumulative exposure index (CEI) was created by summing the product of exposure probability and average level for each year of exposure [49]. For exposure to formaldehyde, CEI was created by summing the product of exposure probability, intensity, and frequency for each year of exposure [50]. CEI were categorized into four categories: never exposed, low, medium and high exposure, according to the percentiles of the distribution among exposed participants (low: < 50th, medium: 50th–90th, high: > 90th) [49].

For the analysis using self-reported chemicals exposure, seven potentially neurotoxic chemical product groups were identified in the self-administered questionnaire: trichloroethylene, white spirit (mineral spirits), cellulosic diluent, paints and varnishes, inks and dyes, pesticides (weed killers, insecticides, fungicides) and formaldehyde. Participants reported if they were exposed to these chemicals during their working life.

## Co-exposure

Two exposure variables to biomechanical risk factors and chemicals were created: one with chemicals estimated by JEM and the other with self-reported chemicals. Both variables had four categories:

• No exposure: no exposure to biomechanical risk factors and no exposure to chemicals,

• Chemical exposure only: only exposed to chemicals,

• Biomechanical exposure only: only exposed to biomechanical risk factors,

• Co-exposure: exposed to both biomechanical risk factors and chemicals.

### Covariables

Age, body mass index (BMI), alcohol use disorders (Alcohol Use Disorders Identification Test, AUDIT-C) and effort–reward imbalance (ERI) were included as main covariates. Diabetes mellitus and/or rheumatoid arthritis (with or without prescription medication) were grouped due to the small number of cases, and considered as a covariate. These covariates were chosen because they were known to be risk factors for CTS [13,6,51,52].

### Statistical analysis

Two analyses were performed: one with lifetime chemical exposure estimated with JEMs and one with lifetime self-reported chemical exposure.

Multiple imputation was performed to handle missing data for covariates (BMI, alcohol use disorders, ERI and diabetes mellitus and/or rheumatoid arthritis) and occupational category with twelve imputed data sets.

Descriptive statistics were performed to describe the study populations. Multivariate logistic regression models were performed to assess the association between co-exposure and CTS. Models were adjusted for age, BMI, diabetes mellitus and/or rheumatoid arthritis, current alcohol consumption and effort-reward imbalance and stratified by gender [53,54]. Interactions between co-exposure and age were tested. Sensitivity analyses included two additional analyses: (1) study of the associations in a subpopulation restricted to low-grade white-collar workers and blue-collar workers and (2) study of the associations using different thresholds: > 2 hours/day for repetitive movements and medium and high to chemical exposure according to cumulative exposure index (vs no and low exposure).

All analyses were performed using R software (version 4.4.1, packages gtsummary, flextable, questionr, forestplot, plyr, stringr, grDevices, UpSetR, ggplot2, ComplexUpset, labelled, dplyr, mice, miceafter and naniar). Results are presented as Odds ratio (OR) with their 95% confidence intervals (95% CI). A p value <0.05 was considered as statistically significant.

## Results

At the time of analysis, data were available for 184,055 participants in the CONSTANCES cohort, including 52,555 who met our inclusion criteria (S1 Fig).

### Lifetime chemical exposure estimated by JEM

Among the 40,889 participants eligible for analysis with lifetime chemical exposure estimated by JEM (18,940 men and 21,949 women), 1.6% [95% CI 1.4–1.8] of men (n = 304) and 2.6% [95% CI 2.4–2.8] of women (n = 564) reported incident CTS at follow-up (p < 0.001).

The median number of jobs held during the career was 3 (IQR 2–5) for men and 3 (IQR 2–4) for women. The median duration of current job was 8 years (IQR 4–16) for men and 9 years (IQR 4–16) for women. Women were more exposed to biomechanical risk factors than men (32.6% vs. 28.0% respectively, p < 0.001). In detail, women reported more high physical demand (21.1% vs. 19.6% respectively, p < 0.001) repetitive movements (11.3% vs. 8.2%, p < 0.001) and repetitive pinching (4.5% vs. 3.6%, p < 0.001), while men were more exposed to awkward wrist postures (7.4% vs. 6.4%, p < 0.001) and hand-transmitted vibrations (3.4% vs. 0.9%, p < 0.001). Men were more exposed to chemicals during their working life than women (14.9% vs. 13.1% respectively, p < 0.001). In detail, men were more exposed to chlorinated solvents (10.4% vs. 2.7%, p < 0.001) and less exposed to formaldehyde (6.4% and 11.5%, p < 0.001). Co-exposure to biomechanical risk factors and chemicals concerned 7.2% of men and 7.6% of women (p < 0.001, Table 1).

More precisely, among men, the most frequent exposures were single exposures: 8.6% were exposed only to high physical perceived exertion, 4.9% only to chlorinated solvents, 3.4% only to repetitive movements and 1.9% only to formaldehyde (S2 Fig). Among women, the most frequent exposures were also single exposures: 10.1% were exposed only

**Table 1. Baseline characteristics of workers (occupational chemical exposure assessed by job-exposure matrices) in men (n = 18,940) and women (n = 21,949).**

| | Men, N = 18,940 n (%) | Women, N = 21,949 n (%) | p-value |
|---|---|---|---|
| **Self- reported carpal tunnel syndrome** | 304 (1.6) | 564 (2.6) | <0.001[a] |
| **Age 45 or over (years)** | 8,235 (43.5) | 9,011 (41.1) | <0.001[a] |
| **Diabetes and/or rheumatoid arthritis** | 202 (1.1) | 133 (0.6) | <0.001[a] |
| **Body mass index (BMI)** | | | <0.001[a] |
| Lean/normal, < 25 kg/m$^2$ | 10,671 (56.3) | 15,700 (71.5) | |
| Overweight, [25–30 kg/m$^2$] | 6,688 (35.3) | 4,435 (20.2) | |
| Obesity, ≥ 30 kg/m$^2$ | 1,581 (8.4) | 1,814 (8.3) | |
| **Alcohol consumption** | | | <0.001[a] |
| Consumption without risk | 6,074 (32.1) | 12,880 (58.7) | |
| Abstinence | 821 (4.3) | 1,820 (8.3) | |
| Consumption with low risk | 10,364 (54.7) | 6,490 (29.6) | |
| Alcohol use disorders | 1,681 (8.9) | 759 (3.5) | |
| **Occupational category** | | | |
| Farmers | 7 (0.0) | 6 (0.0) | |
| Craftsmen, salesmen and managers | 281 (1.5) | 133 (0.6) | |
| Professionals | 7,489 (39.5) | 5,068 (23.1) | |
| Technicians and associate professionals | 7,392 (39.0) | 11,345 (51.7) | |
| Lower-level white-collar workers | 1,717 (9.1) | 5,069 (23.1) | |
| Blue collar workers | 2,054 (10.8) | 328 (1.5) | |
| **Effort-reward imbalance ratio >1** | 8,569 (45.2) | 11,059 (50.4) | |
| **Recent biomechanical exposure** | 5,306 (28.0) | 7,165 (32.6) | <0.001[a] |
| High physical perceived exertion (RPE > 12) | 3,708 (19.6) | 4,628 (21.1) | <0.001[a] |
| Repetitive hand movements (>4h/day) | 1,545 (8.2) | 2,489 (11.3) | <0.001[a] |
| Awkward wrist postures (≥ 2h/day) | 1,397 (7.4) | 1,398 (6.4) | <0.001[a] |
| Repetitive pinching (> 4h/day) | 690 (3.6) | 990 (4.5) | <0.001[a] |
| Hand-transmitted vibrations (≥ 2h/day) | 638 (3.4) | 192 (0.9) | <0.001[a] |
| **Lifetime occupational chemical exposure** | 2,829 (14.9) | 2,885 (13.1) | <0.001[a] |
| At least one chlorinated solvent | 1,968 (10.4) | 585 (2.7) | <0.001[a] |
| Duration of exposure | | | 0.300[b] |
| Mean (Standard deviation) | 9.4 (8.2) | 10.3 (9.5) | |
| Median (Interquartile range) | 7.0 (3.0-13.0) | 7.0 (3.0-14.0) | |
| Min-Max | 1.0-44.0 | 1.0-,41.0 | |
| Cumulative Exposure Index | | | <0.001[a] |
| Non exposed | 16,972 (89.6) | 21,364 (97.3) | |
| Low | 619 (3.3) | 241 (1.1) | |
| Medium | 656 (3.5) | 199 (0.9) | |
| High | 693 (3.7) | 145 (0.7) | |
| Formaldehyde | 1,216 (6.4) | 2,531 (11.5) | <0.001[a] |
| Duration of exposure | | | 0.060[b] |
| Mean (Standard deviation) | 10.9 (9.5) | 11.5 (9.6) | |
| Median (Interquartile range) | 8.0 (3.0-16.0) | 9.0 (4.0-17.0) | |
| Min-Max | 1.0-42.0 | 1.0-40.0 | |
| Cumulative Exposure Index | | | <0.001[a] |
| Non exposed | 17,724 (93.6) | 19,418 (88.5) | |
| Low | 363 (1.9) | 899 (4.1) | |

*(Continued)*

**Table 1.** (Continued)

| | Men, N = 18,940 n (%) | Women, N = 21,949 n (%) | p-value |
|---|---|---|---|
| Medium | 399 (2.1) | 855 (3.9) | |
| High | 454 (2.4) | 777 (3.5) | |
| **Cumulated exposure to lifetime occupational chemical exposure** | | | <0.001ᵃ |
| No exposed | 16,111 (85.1) | 19,064 (86.9) | |
| Exposed only to at least one chlorinated solvent or only to formaldehyde | 2,474 (13.1) | 2,654 (12.1) | |
| Exposed to at least one chlorinated solvent and formaldehyde | 355 (1.9) | 231 (1.1) | |
| **Biomechanical-chemical exposure** | | | <0.001ᵃ |
| No exposure | 12,171 (64.3) | 13,572 (61.8) | |
| Chemical exposure only | 1,463 (7.7) | 1,212 (5.5) | |
| Biomechanical exposure only | 3,940 (20.8) | 5,492 (25.0) | |
| Co-exposure | 1,366 (7.2) | 1,673 (7.6) | |

[a]Chi-squared test.

[b]Wilcoxon rank sum test.

The distribution of BMI, alcohol use disorders, ERI and diabetes mellitus and/or rheumatoid arthritis was described after multiple imputation.

to high physical perceived exertion, 5.7% only to formaldehyde, 4.1% only to repetitive movements; 3.8% of women were exposed to high physical perceived exertion and formaldehyde (S3 Fig).

Univariate analysis is shown in S4 Table. In the multivariable analysis, a statistically significant association (p < 0.001) was found for both genders between CTS and exposure to biomechanical risk factors and chemicals. The adjusted OR in the biomechanical exposure group were 2.02 [1.55–2.63] in men and 1.46 [1.20–1.77] in women, and the adjusted OR in the co-exposure group were 2.37 [1.67–3.38] in men and 2.05 [1.58–2.66] in women, compared to non-exposed workers (Fig 1). Age did not show any interaction with co-exposure.

## Self-reported lifetime chemical exposure

Among the 46,809 participants eligible for analysis using self-reported lifetime chemical exposure (22,751 men and 24,058 women), 1.6% [95% CI 1.5–1.8] of men (n = 371) and 2.5% [95% CI 2.3–2.7] of women (n = 611) reported incident CTS at follow-up (p < 0.001).

The median number of jobs and the median duration of current job were identical to the JEM analysis. Women were more exposed to biomechanical risk factors than men (33.2% vs. 29.3% respectively, p < 0.001) and men were more exposed to chemicals during their working life than women (16.4% vs. 7.0%, p < 0.001). Co-exposure to biomechanical risk factors and chemicals concerned 8.2% of men and 3.5% of women (p < 0.001, Table 2) [55]. Among men, the most frequent exposures were single exposures: 8.7% were exposed only to high physical perceived exertion and 3.3% only to repetitive movements (S5 Fig). Among women, the most frequent exposures were: exposed only to high physical perceived exertion (13.0%), exposed only to repetitive movements (5.9%) and exposed to high physical perceived exertion and repetitive movements (2.2%) (S6 Fig).

Univariate analysis is shown in S4 Table. Similar to the JEM, a statistically significant association (p < 0.001) was found between CTS and exposure to biomechanical risk factors and chemicals for both genders. The adjusted OR in the biomechanical exposure group were 1.92 [1.50–2.46] in men and 1.53 [1.29–1.82] in women (ref: non-exposed workers). The adjusted OR in the co-exposure group were higher than in the analysis with JEM: 2.97 [2.22–3.97] in men and 2.49 [1.80–3.44] in women, compared to non-exposed workers (Fig 2). Age did not show any interaction with co-exposure.

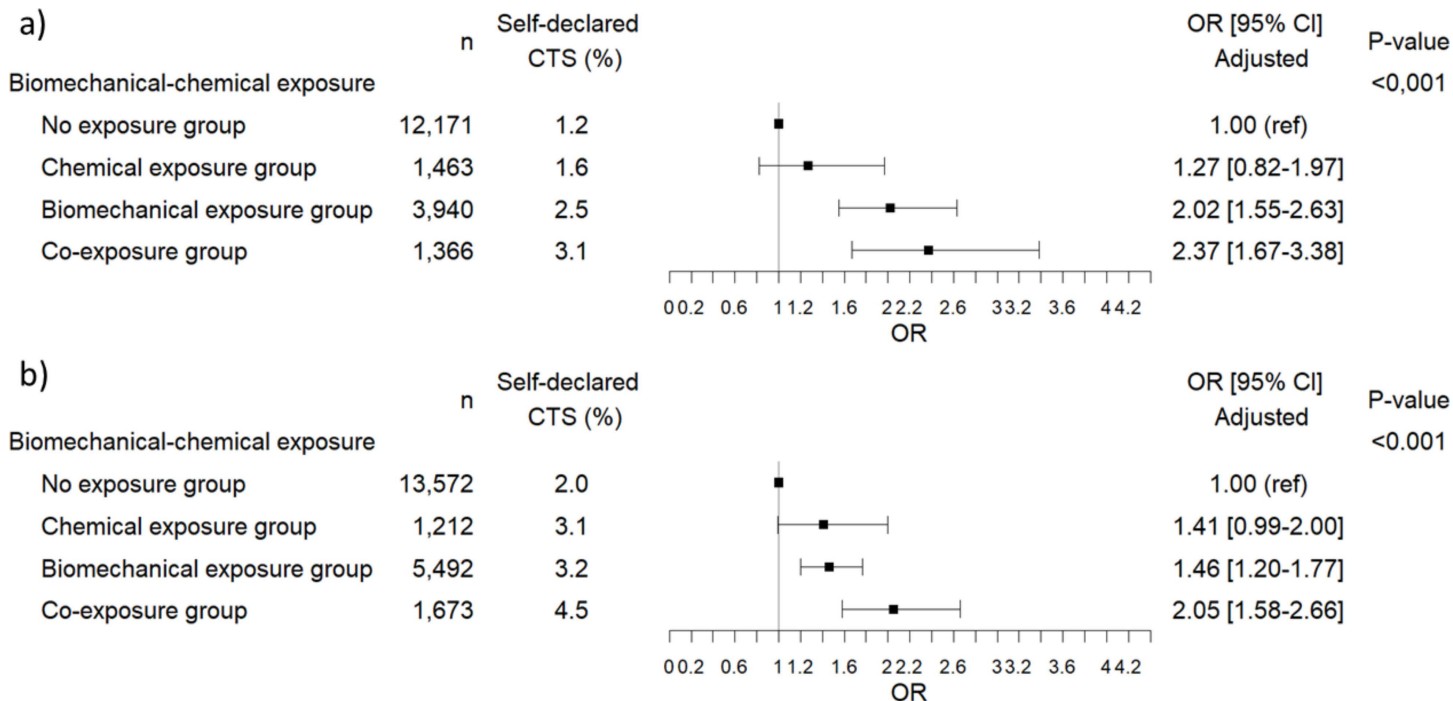

**Fig 1. Multivariate risk models for CTS in a) men (n = 18,940) and b) women (n = 21,949) after adjustment for covariates (age, BMI, diabetes mellitus and/or rheumatoid arthritis, current alcohol consumption and effort-reward imbalance) with lifetime chemical exposure estimated by JEM.**

## Sensitivity analyses

When the population was restricted to low-grade white-collar workers and blue-collar workers, the results were globally similar in women and less significant in men (S7 and S8 Figs). The sensitivity analysis with different thresholds showed similar results (S9 Fig).

## Discussion

Our previous study on a sample of French professionally active adults enrolled in the Constances cohort had showed an higher association between self-reported CTS and co-exposure to biomechanical risk factors and chemicals in both genders after adjustment for personal and medical factors and effort–reward imbalance ratio [15]. The present study, which involved a larger sample, confirmed the associations between co-exposure and incident CTS, by using two different methods to assess chemical exposure (JEMs and self-report).

Little epidemiological data is available on the association between chemical exposures and CTS in spite of their potential neurotoxicity. The association for the co-exposure group may reflect synergic effects of biomechanical stressors and chemicals. Mechanistically, pathophysiological studies show that increased intra-canal pressure (ICP) secondary to wrist movements and external compression at the carpal tunnel represents the main mechanism behind the intraneural vascular disruption and the median nerve damage (peripheral biomechanical component) [56]. Central sensory-motor mechanisms secondary to abnormalities in sensory feedback and motor control of the hand have more recently been highlighted (central sensory-motor component) [57]. Concerning the association for chemical and biomechanical co-exposures, the median nerve at the level of the carpal tunnel is particularly exposed to biomechanical stresses due to wrist mobility. Two

**Table 2. Baseline characteristics of workers (occupational chemical exposure assessed by self-reported) in men (n = 22,751) and women (n = 24,058).**

| | Men, N = 22,751 n (%) | Women, N = 24,058 n (%) | p-value[a] |
|---|---|---|---|
| **Self- reported carpal tunnel syndrome** | 371 (1.6) | 611 (2.5) | <0.001 |
| **Age 45 or over (years)** | 9,796 (43.1) | 9,620 (40.0) | <0.001 |
| **Diabetes and/or rheumatoid arthritis** | 250 (1.1) | 141 (0.5) | <0.001 |
| **Body mass index (BMI)** | | | <0.001 |
| Lean/normal, < 25 kg/m$^2$ | 12,764 (56.1) | 17,144 (71.3) | |
| Overweight, [25–30 kg/m$^2$] | 8,054 (35.4) | 4,886 (20.3) | |
| Obesity, ≥ 30 kg/m$^2$ | 1,933 (8.5) | 2,028 (8.4) | |
| **Alcohol consumption** | | | <0.001 |
| Consumption without risk | 7,261 (31.9) | 14,097 (58.6) | |
| Abstinence | 1,020 (4.5) | 1,990 (8.3) | |
| Consumption with low risk | 12,449 (54.7) | 7,150 (29.7) | |
| Alcohol use disorders | 2,021 (8.9) | 821 (3.4) | |
| **Effort-reward imbalance ratio >1** | 10,353 (45.5) | 12,063 (50.1) | <0.001 |
| **Occupational category** | | | <0.001 |
| Farmers | 10 (0.0) | 7 (0.0) | |
| Craftsmen, salesmen and managers | 335 (1.5) | 150 (0.6) | |
| Professionals | 8,936 (39.3) | 5,776 (24.0) | |
| Technicians and associate professionals | 8,526 (37.5) | 12,076 (50.2) | |
| Lower-level white-collar workers | 1,906 (8.4) | 5,475 (22.8) | |
| Blue collar workers | 3,038 (13.4) | 574 (2.4) | |
| **Missing** | | | |
| **Recent biomechanical exposure** | 6,659 (29.3) | 7,998 (33.2) | <0.001 |
| High physical perceived exertion (RPE > 12) | 4,771 (21.0) | 5,206 (21.6) | 0.077 |
| Repetitive hand movements (>4h/day) | 1,950 (8.6) | 2,839 (11.8) | <0.001 |
| Awkward wrist postures (≥ 2h/day) | 1,846 (8.1) | 1,609 (6.7) | <0.001 |
| Repetitive pinching (> 4h/day) | 833 (3.7) | 1,100 (4.6) | <0.001 |
| Hand-transmitted vibrations (≥ 2h/day) | 865 (3.8) | 235 (1.0) | <0.001 |
| **Lifetime occupational chemical exposure** | 3,725 (16.4) | 1,686 (7.0) | <0.001 |
| Trichloroethylene | 1,122 (4.9) | 251 (1.0) | <0.001 |
| White (mineral) spirit | 1,326 (5.8) | 265 (1.1) | <0.001 |
| Cellulosic diluent | 609 (2.7) | 98 (0.4) | <0.001 |
| Paints, varnishes | 1,489 (6.5) | 462 (1.9) | <0.001 |
| Inks, dyes | 406 (1.8) | 293 (1.2) | <0.001 |
| Pesticides | 907 (4.0) | 327 (1.4) | <0.001 |
| Formaldehyde | 449 (2.0) | 625 (2.6) | <0.001 |
| **Biomechanical-chemical exposure** | | | <0.001 |
| No exposure | 14,239 (62.6) | 15,213 (63.2) | |
| Chemical exposure only | 1,853 (8.1) | 847 (3.5) | |
| Biomechanical exposure only | 4,787 (21.0) | 7,159 (29.8) | |
| Co-exposure | 1,872 (8.2) | 839 (3.5) | |

[a]Chi-squared test.

The distribution of BMI, alcohol use disorders, ERI and diabetes mellitus and/or rheumatoid arthritis was described after multiple imputation.

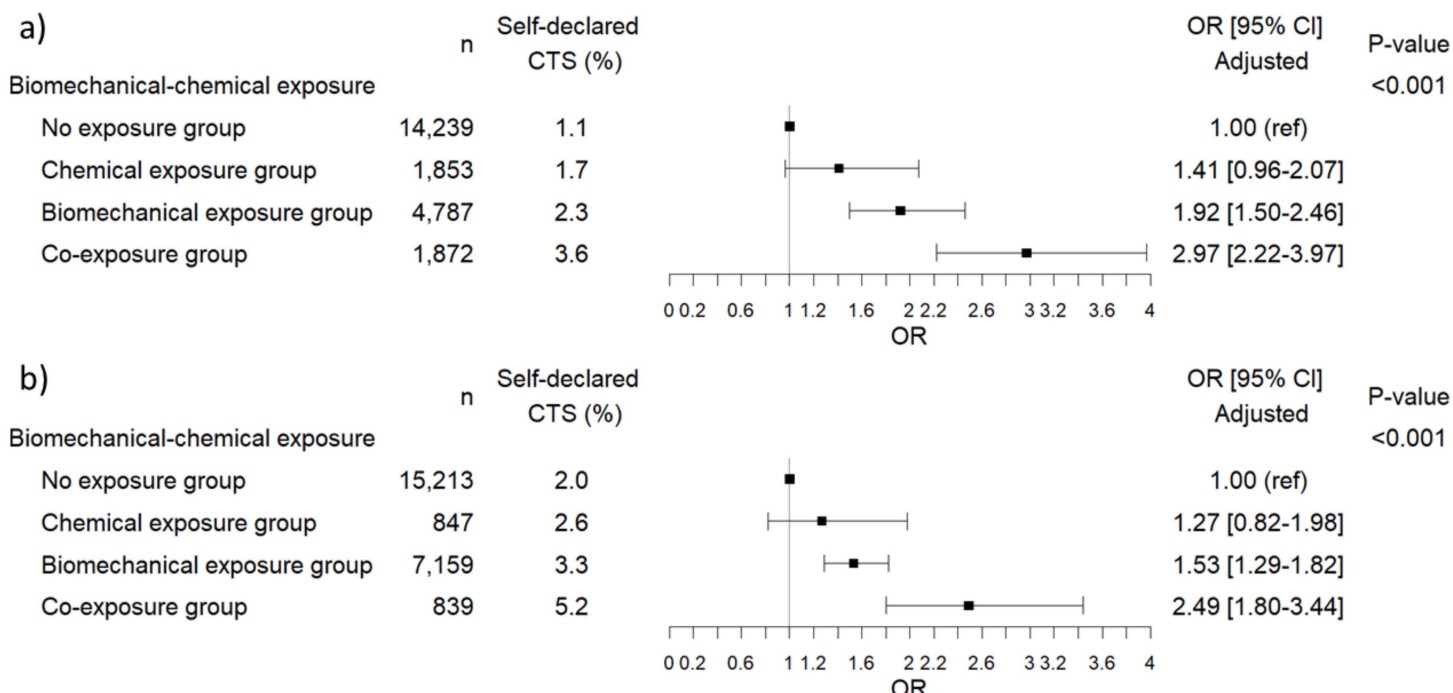

**Fig 2. Multivariate risk models for CTS in a) men (n = 20,877) and b) women (n = 21,291) after adjustment for covariates (age, BMI, diabetes mellitus and/or rheumatoid arthritis, current alcohol consumption and effort-reward imbalance) with self-reported lifetime exposure to chemicals.**

putative mechanisms, peripheral and central, may be involved in the combined effects of neurotoxic chemical and biomechanical exposures on the risk of CTS.

First, systemic peripheral nerve damage (peripheral mechanism), secondary to neurotoxic exposure, could induce diffuse nerve microdamage making the median nerve more sensitive to compression in the carpal tunnel secondary to biomechanical stresses (concept of "double crush syndrome" induced by chemicals) [32–34]. Second, central nervous lesions of neurotoxic origin lead to functional and/or structural alterations of the CNS [19,22,25,35,36,50,58] (Central mechanism), in particular in the sensorimotor zones likely to increase the effects of CTS by (i) impairing sensorimotor coordination and disrupting the biomechanical efficiency of professional movements, and (ii) accentuating the decrease in dexterity induced by anomalies in haptic sensory feedback during CTS [59–61]. These mechanisms could lead to an increase in the biomechanical stresses applied to the wrist, and thus create a vicious circle likely to accentuate the risk of CTS in co-exposed workers. However, we cannot rule out that chemical exposure may also be a marker for greater overuse of hand strength at work. Given the rather rudimentary assessment of wrist exposure, it may be that part of the effect of combined exposure is linked to greater biomechanical exposure at the wrist. Indeed, the proportion of blue-collar workers was higher among co-exposed individuals, and it might be expected that blue-collar workers would be more exposed to chemicals and use their hands at higher intensities than employees reporting the same level of hand use. Odds ratios were broadly similar when the study population was restricted to lower-level white-collar and blue-collar workers, but the associations were statistically significant only in women.

The prospective design is one of the major strengths of the study. The Constances cohort includes a large sample of employed men and women of various age groups from different regions in France, covering a variety of occupational groups [37]. Lifetime chemical exposure was evaluated by two methods: JEM and self-report. Both methods found an association between co-exposure to biomechanical risk factors and chemicals and CTS. In addition, the main potential

personal and medical risk factors for CTS were taken into account (e.g., age, BMI, diabetes mellitus, arthritis, and alcohol consumption) [4,11]. Analyses were stratified according to gender following the recommendations for the study of musculoskeletal disorders [53,54]. The differences in exposure frequency between men and women could be explained by the fact that women and men do not occupy the same jobs. In addition, even within the same occupation, several studies have shown that exposure differs between men and women due to the gendered distribution of tasks, missions and workspaces [62].

There are several limitations to this study. Despite its size, the study sample does not represent the whole structure of occupations in France, as self-employed workers and farmers are not included in the cohort. There are two cohorts set up by Santé publique France (Coset-MSA and Coset-Indépendants) that would allow to study these specific populations [63].

Epidemiological studies include several definitions of CTS [64]. In our study, CTS was reported by participants using a single question (yes/no). We cannot rule out misclassification, as CTS may have been over-reported by workers who believe they have been exposed to occupational factors causing CTS, whether or not their hand symptoms resemble those of CTS. The cohort design did not allow for the use of a symptom algorithm, physical examination or EMG confirmation, which is an important limitation of the study. The working population studied probably did not include the most severe cases of CTS that may have been excluded from the labor market (i.e., the healthy worker effect), which could lead to an underestimate of the prevalence of CTS.

Measuring self-reported occupational exposure is an important limitation of the study. Direct observation or measurement is costly and difficult to apply to large cohorts. In addition, the assessment of biomechanical risk factors was requested for a typical working day and was based on the recommendations of the European criteria document for the relatedness of MSDs [38], and postures were presented in pictorial form to facilitate workers' understanding and increase the validity of posture self-assessment. We cannot exclude that biomechanical and chemical exposures were over-reported by workers reporting work-related CTS, which is a potential source of reverse causality. To limit this bias, we have excluded subjects reporting chronic wrist pain and/or CTS at baseline. Nevertheless, there is no evidence to suggest a differential bias depending on whether workers were exposed to biomechanical factors alone or to biomechanical and chemical factors combined. Moreover, the prevalence of self-reported CTS was of the same order of magnitude depending on whether there was biomechanical exposure alone or biomechanical and chemical co-exposure (in analysis with JEM: 2.5% vs 3.1% in men and 3.2% vs 4.5% in women; in analysis with self-report: 2.3% vs 3.6% in men and 3.3% vs 5.2% in women). Although the difference in prevalence was relatively modest, we do not believe that classification bias can explain the results, given the consistency of the results with a subjective assessment and an objective assessment of chemical exposure by JEM.

The list of chemicals was different between the JEM and self-reported analyses, with only trichloroethylene and formaldehyde common to both. Indeed, the self-administered questionnaire does not allow detailed questioning about chemicals (chemical substances, chemical families, complex mixtures and product categories) [17], as workers may be unfamiliar with the chemicals to which they have been exposed during their working careers. Moreover, lifetime self-report was assessed using a single ever/never question and memory bias cannot be ruled out. JEM has the advantage of providing a more accurate estimate of chemical exposure, especially for past exposures. However, by construction, JEM considers that people within the same job code and a given period have the same exposure. Misclassification of exposure is expected to be non-differential with respect to the health outcome, the true association might be stronger than observed. In addition, some known neurotoxic chemicals, such as lead, mercury, n-hexane, were not available in JEM. JEMs developed by Santé publique France for petroleum and oxygenated solvents will soon be updated with the same nomenclatures of jobs as Constances, and analyses taking these solvents into account could be carried out in the future. Given the exposure of workers to a mixture of solvents rather than to a given type of solvent, it cannot be ruled out that workers exposed to chlorinated solvents were also exposed to aliphatic or aromatic solvents [16]. Given the large size of the cohort (over 200,000 adults) and the global nature of the cohort, questions deemed relevant to the study of CTS could

not be asked, such as ergonomic measures in the workplace and the wearing of protective equipment. We chose covariates based on the literature [13,6,51,52], and opted to not include tobacco consumption [51]. Physical activity would have also been relevant. However, the physical activity practiced was not indicated in the questionnaire, and some sports, such as tennis or golf have an impact on CTS, while others have no impact. Endocrine disorders such as hyperthyroidism and hypothyroidism were not available in the questionnaires at baseline. In addition, there is no validated algorithm for identifying these disorders in French administrative health care database. Consequently, these disorders could not be used as covariates.

In conclusion, this large prospective study showed an association between self-reported CTS and co-exposure to both occupational biomechanical risk factors and chemicals, using JEM and self-report methods. These results are consistent with those of our previous studies in the same cohort with self-reported chemical exposure [15], and in a cohort of French farmers, although, we lack mechanistic evidence, provide further support for the potential effects of chemical exposure on the risk of CTS in workers exposed to biomechanical risk factors. Access to French administrative health care database will enable us to study associations between surgically treated CTS and occupational co-exposures in a future study. It would also be relevant to study other chemical exposures (petroleum and oxygenated solvents), as well as biomechanical exposures by considering a more objective measure such as JEM.

## Supporting information

**S1 Fig. Flow-chart of the study population.**
(TIF)

**S2 Fig. Upset plot of biomechanical (self-reported) and chemical exposure (job- exposure matrix) in men (Constances cohort, France; frequency of less than 30 are not shown).** Note: 8.6% of men were exposed only to high physical perceived exertion (n = 1,625). 1.4% of men were exposed to high physical perceived exertion and formaldehyde (n = 258).
(TIF)

**S3 Fig. Upset plot of biomechanical (self-reported) and chemical exposure (job- exposure matrix) in women (Constances cohort, France; frequency of less than 30 are not shown).** Note: 10.1% of women were exposed only to high physical perceived exertion (n = 2,209). 3.8% of women were exposed to high physical perceived exertion and formaldehyde (n = 827).
(TIF)

**S4 Table. Univariate risk models for CTS in men and women.**
(DOCX)

**S5 Fig. Upset plot of biomechanical and chemical exposure (self-reported) matrix in men (Constances cohort, France; frequency of less than 30 are not shown).** Note: 8.7% of men were exposed only to high physical perceived exertion (n = 1,990). 1.3% of men were exposed to high physical perceived exertion and repetitive hand movements (n = 299).
(TIF)

**S6 Fig. Upset plot of biomechanical and chemical exposure (self- reported) matrix in women (Constances cohort, France; frequency of less than 30 are not shown).** Note: 13.0% of women were exposed only to high physical perceived exertion (n = 3,121). 2.2% of women were exposed to high physical perceived exertion and repetitive hand movements (n = 535).
(TIF)

**S7 Fig. Multivariate risk models for CTS in a) men and b) women after adjustment for covariates (age, BMI, diabetes mellitus and/or rheumatoid arthritis, current alcohol consumption and effort-reward imbalance) with lifetime chemical exposure estimated by JEM in low-grade white-collar workers and blue-collar workers.** This is the S6 Fig legend.
(TIF)

**S8 Fig. Multivariate risk models for CTS in a) men and b) women after adjustment for covariates (age, BMI, diabetes mellitus and/or rheumatoid arthritis, current alcohol consumption and effort-reward imbalance) with self-reported lifetime exposure to chemicals in low-grade white-collar workers and blue-collar workers.**
(TIF)

**S9 Fig. Multivariate risk models for CTS in a) men and b) women after adjustment for covariates (age, BMI, diabetes mellitus and/or rheumatoid arthritis, current alcohol consumption and effort-reward imbalance) with lifetime chemical exposure estimated by JEM using different thresholds.** Thresholds: > 2 hours/day for repetitive movements and medium and high exposure to chemical exposure, according to cumulative exposure index (vs no and low exposure).
(TIF)

## Acknowledgments

The authors thank the team of the "Population-based cohorts unit" (Cohortes en population) that designed and manages the Constances cohort study. They also thank the French national health insurance ("Caisse nationale d'assurance maladie", Cnam) and its Health screening centres ("Centres d'examens de santé"), which are collecting a large part of the data, as well as the French national old-age insurance ("Caisse nationale d'assurance vieillesse", Cnav) for its contribution to the constitution of the cohort, and ClinSearch, Asqualab and Eurocell, which are conducting the data quality control. The authors thank the members of the Matgéné working group from Santé publique France for providing job-exposure matrices.

## Author contributions

**Conceptualization:** Julie Bodin, Yves Roquelaure.

**Data curation:** Marcel Goldberg, Marie Zins.

**Formal analysis:** Julie Bodin, Clémence Rapicault, Fabien Gilbert.

**Funding acquisition:** Yves Roquelaure.

**Supervision:** Yves Roquelaure.

**Validation:** Julie Bodin, Clémence Rapicault.

**Writing – original draft:** Julie Bodin.

**Writing – review & editing:** Julie Bodin, Clémence Rapicault, Alexis Descatha, Marc Fadel, Fabien Gilbert, Bradley Evanoff, Nathalie Bonvallot, Marcel Goldberg, Marie Zins, Yves Roquelaure.

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
