## [Decision Letter · Decision Letter 0]

12 Feb 2025

Dear Dr. Bodin,

We look forward to receiving your revised manuscript.

Kind regards,

Giulia Squillacioti

Academic Editor

PLOS ONE

Journal requirements:   When submitting your revision, we need you to address these additional requirements. 1. Please ensure that your manuscript meets PLOS ONE's style requirements, including those for file naming. The PLOS ONE style templates can be found at https://journals.plos.org/plosone/s/file?id=wjVg/PLOSOne_formatting_sample_main_body.pdf and https://journals.plos.org/plosone/s/file?id=ba62/PLOSOne_formatting_sample_title_authors_affiliations.pdf. 2. We note that the grant information you provided in the ‘Funding Information’ and ‘Financial Disclosure’ sections do not match.  When you resubmit, please ensure that you provide the correct grant numbers for the awards you received for your study in the ‘Funding Information’ section. 3. Thank you for uploading your study's underlying data set. Unfortunately, the repository you have noted in your Data Availability statement does not qualify as an acceptable data repository according to PLOS's standards. At this time, please upload the minimal data set necessary to replicate your study's findings to a stable, public repository (such as figshare or Dryad) and provide us with the relevant URLs, DOIs, or accession numbers that may be used to access these data. For a list of recommended repositories and additional information on PLOS standards for data deposition, please see https://journals.plos.org/plosone/s/recommended-repositories.

Reviewers' comments:

Reviewer's Responses to Questions

**Comments to the Author**

1. Is the manuscript technically sound, and do the data support the conclusions?

Reviewer #1: Yes

Reviewer #2: Partly

2. Has the statistical analysis been performed appropriately and rigorously?

Reviewer #1: Yes

Reviewer #2: No

3. Have the authors made all data underlying the findings in their manuscript fully available?

Reviewer #1: Yes

Reviewer #2: Yes

4. Is the manuscript presented in an intelligible fashion and written in standard English?

Reviewer #1: Yes

Reviewer #2: Yes

Reviewer #1: This manuscript addresses an important and timely topic in occupational health, focusing on the relationship between biomechanical and chemical co-exposures and the development of carpal tunnel syndrome (CTS). The study leverages the CONSTANCES cohort, a unique and robust dataset that integrates comprehensive occupational exposure histories, detailed self-reported information, and job-exposure matrices (JEM). This combination of data sources provides a rare opportunity to explore the complex interactions between multiple workplace exposures and their cumulative effects on workers' health.

What sets this research apart is its focus on the co-exposure paradigm—an underexplored yet critical area in occupational epidemiology. While prior studies have often isolated biomechanical and chemical risks, this work seeks to understand their combined impact, offering a more holistic view of workplace hazards. The findings have the potential to inform targeted interventions and preventive strategies, thereby improving worker health and safety.

By addressing key knowledge gaps and utilizing an innovative methodological approach, this research contributes significantly to the broader understanding of multi-exposure scenarios and their implications for occupational health outcomes. However, as with any study, there are areas for refinement to enhance the clarity, rigor, and practical relevance of the findings. This review highlights both the strengths and the areas needing improvement, providing constructive feedback to further strengthen the manuscript.

Conceptual and Theoretical Framework

Justification of Hypotheses:The hypothesis that biomechanical and neurotoxic chemical exposures synergistically increase the risk of carpal tunnel syndrome (CTS) needs stronger theoretical backing. The concept of the 'double-crush syndrome' is mentioned but not thoroughly supported with detailed mechanistic studies or references. Recommend including recent studies or reviews on neurotoxicity and its impact on peripheral nerve susceptibility to mechanical compression.

Integration of Neurotoxic Chemical Evidence:The paper lacks detailed discussion of the specific neurotoxic pathways of chlorinated solvents and formaldehyde and their relevance to CTS development. Expand on known mechanisms (e.g., inflammation, reactive oxygen species overproduction) and provide references to strengthen the link between chemical exposure and nerve damage.

Methodology

Self-Reported Bias:Reliance on self-reported CTS and chemical exposure introduces recall and reporting bias, potentially overestimating associations. Suggest validating self-reports with clinical data, such as medical records or surgery databases, and conducting sensitivity analyses to estimate the impact of misclassification. Further describing any methods employed (if any) to minimize these biases would be helpful.

Limited Chemical Scope:The exclusion of other known neurotoxic chemicals (e.g., heavy metals, polycyclic aromatic hydrocarbons) is a significant limitation. Justify this choice in the methods text or explicitly acknowledge it as a limitation in the discussion section.

Gender Stratification Rationale:Gender differences in exposure and CTS risk are presented but not deeply analyzed. Explore possible reasons for these differences, such as occupational roles, task-specific exposures, or physiological factors.

Potential Confounders:Effort-reward imbalance and alcohol consumption are included as covariates, but other factors (e.g., psychosocial stress, exposure to additional workplace hazards) could confound the results. Consider conducting additional analyses or cite literature supporting the adequacy of the chosen covariates or providing more detailed rationale why they were not considered.

Statistical Model Details:The methodology section does not adequately explain key statistical choices, such as the selection of logistic regression, handling of interaction terms, and treatment of missing data. Include a rationale for these decisions and clarify how missing data were addressed (e.g., multiple imputation or exclusion). Additionally, assessment of potential confounders and the decision process for those should be described further.

Threshold Definitions for Exposure:The thresholds for biomechanical exposure (e.g., repetitive movements for >4 hours/day) and chemical exposure categories appear arbitrary. Consider conducting and reporting sensitivity analyses to evaluate whether changing these thresholds affects results.

Discussion and Conclusion

Causality Limitations:The results are presented as though they imply causation, despite the inherent limitations of observational studies. Emphasize the associative nature of the findings and discuss potential reverse causality (e.g., pre-existing conditions influencing exposure reports).

Overemphasis on Findings:The conclusion overstates the implications of the findings by implying a causal relationship. Adjust language to reflect the observational design and highlight areas where further research is needed to confirm causality.

Study Population Representation:The exclusion of self-employed workers, farmers, and other groups limits the generalizability of the findings. Address this limitation in the discussion and explore the potential impact on results.

Future Research Suggestions:Provide specific and actionable recommendations for future studies, such as using clinical CTS diagnoses, exploring additional chemical exposures, or utilizing objective biomechanical exposure assessments.

Ethical and Practical Considerations

Ethical Concerns in Data Use:The data accessibility restrictions, while necessary for participant confidentiality, hinder replication efforts. Include clear instructions for researchers on how to obtain the necessary permissions and datasets.

Typos and Grammar

Abstract:

- Improper sentence structure: 'cohort that included, between 2012 and 2019, a randomly selected sample of adults.'

- Awkward phrasing: 'Lifetime occupational exposure to chemicals was assessed using both methods: JEM and a self-administered questionnaire completed at inclusion.'

Methods:

- Repetitive statement: 'All volunteers signed a written consent form for their participation in CONSTANCES.'

Results:

- Inconsistent spacing in statistical reporting: 'OR = 3.07 [2.28-4.08].'

Discussion:

- Redundant sentence: 'This finding confirms previous results in the same cohort.'

Reviewer #2: General Comments

The study evaluates the association between occupational co-exposure to biomechanical risk factors and potentially neurotoxic chemicals with carpal tunnel syndrome (CTS) in a cohort of French workers. It is relevant because it addresses one of the most common entrapment neuropathies in workers exposed to biomechanical and chemical factors. Additionally, the use of a large cohort significantly enhances the consistency of the findings and allows for the comparison of different approaches to estimate chemical exposure in the workplace. However, the self-reported diagnosis of CTS may introduce bias. Furthermore, the study does not analyze the intensity or duration of exposure, and there are potential unconsidered confounding factors. Additionally, the data analysis does not specify whether it was adjusted for the type of random sampling used.

Regarding the writing style, there are spelling errors in all sections of the manuscript that need to be corrected.

Introduction

The introduction presents the problem with appropriate wording and is supported by scientific literature from the past 15 years. No major observations were found, except for some spelling errors.

Methods

The self-reported outcome is the primary limitation of this study. The authors should justify why they chose this approach.

What was the rationale for assessing the outcome at 12 months? While CTS can develop within months or a few years (depending on the high biomechanical demand), most workers develop it over a period exceeding five years. The follow-up period needs to be justified, and if relevant, considered as a potential limitation.

Regarding covariates, important confounders were not included, such as: Duration of exposure to biomechanical or chemical agents, interventions to reduce CTS risk, such as ergonomic measures, active breaks, or protective equipment, physical activity outside the workplace, smoking habits, and individual factors, such as endocrine disorders or the use of medications affecting nerve function.

These variables should be explicitly detailed in the limitations section.

The statistical analysis does not indicate whether the calculations were adjusted for the random sampling method declared in the study. If not accounted for, this could significantly impact the results. Additionally, the authors should clarify whether they employed any strategies to handle missing data.

The study mentions a sex-stratified analysis, but no justification is provided. Stratification by type of occupational activity could provide more specific risk measures. The authors should evaluate this possibility.

The authors used logistic regression models adjusted for confounders; however, given the low frequency of the outcome (CTS), an alternative approach would be to use Firth logistic regression, which corrects the overestimation of odds ratios (ORs).

Results

The authors should indicate whether any workers were excluded during the follow-up period and provide reasons for their exclusion, as Figure S1 does not mention this.

In Table 2, some data are in French, and the translation should be completed. Additionally, the p-values should include superscripts to specify the statistical test used.

Discussion

The authors should explain how exposure misclassification could lead to an underestimation or overestimation of individual exposure and propose strategies to overcome this limitation.

Regarding the outcome (CTS diagnosis), the authors should consider discussing more objective assessment alternatives, such as: validated clinical algorithms, electromyographic testing, and medical records of carpal tunnel release surgery.

Incorporating these aspects would strengthen the validity of the findings and minimize potential biases in CTS classification.

**Do you want your identity to be public for this peer review?** For information about this choice, including consent withdrawal, please see our Privacy Policy

Reviewer #1: **Yes: ** Matthew S. Thiese, PhD, MSPH

Reviewer #2: **Yes: ** Jaime Rosales-Rimache

---

## [Author Response · Author response to Decision Letter 1]

28 Mar 2025

Reviewer #1: This manuscript addresses an important and timely topic in occupational health, focusing on the relationship between biomechanical and chemical co-exposures and the development of carpal tunnel syndrome (CTS). The study leverages the CONSTANCES cohort, a unique and robust dataset that integrates comprehensive occupational exposure histories, detailed self-reported information, and job-exposure matrices (JEM). This combination of data sources provides a rare opportunity to explore the complex interactions between multiple workplace exposures and their cumulative effects on workers' health.

What sets this research apart is its focus on the co-exposure paradigm—an underexplored yet critical area in occupational epidemiology. While prior studies have often isolated biomechanical and chemical risks, this work seeks to understand their combined impact, offering a more holistic view of workplace hazards. The findings have the potential to inform targeted interventions and preventive strategies, thereby improving worker health and safety.

By addressing key knowledge gaps and utilizing an innovative methodological approach, this research contributes significantly to the broader understanding of multi-exposure scenarios and their implications for occupational health outcomes. However, as with any study, there are areas for refinement to enhance the clarity, rigor, and practical relevance of the findings. This review highlights both the strengths and the areas needing improvement, providing constructive feedback to further strengthen the manuscript.

We thank Reviewer #1 for his feedback on our manuscript. We greatly appreciate the time and effort he has put into his review. Our specific responses are provided below.

Conceptual and Theoretical Framework

Justification of Hypotheses:The hypothesis that biomechanical and neurotoxic chemical exposures synergistically increase the risk of carpal tunnel syndrome (CTS) needs stronger theoretical backing. The concept of the 'double-crush syndrome' is mentioned but not thoroughly supported with detailed mechanistic studies or references. Recommend including recent studies or reviews on neurotoxicity and its impact on peripheral nerve susceptibility to mechanical compression.

According to your remark, we have added references on the association between the association between CTS and chemical exposure and we have detailed the justification of the hypotheses in the Introduction section.

“To the best of our knowledge, the impact of chemical exposure on the risk of CTS has rarely been studied despite the potential neurotoxicity of some chemicals [15,26–31]. Ophir et al. [29] reported an increased risk of CTS-like symptoms following subclinical sensory polyneuropathy (affecting mainly the median and sural nerves) in workers exposed to prolonged low-level exposure to organophosphate in rural communities in Israel. However, a case-control study of CTS conducted in the general population of Wisconsin failed to report an association between CTS and chemical exposure after adjustment for the known risk factors [26]

[…]

Systemic peripheral neuropathies (e.g., diabetic polyneuropathy) are known to cause subtle nerve damage, thus rendering the median nerve more vulnerable to compression within the carpal tunnel and against the volar ligament in the case of overexposure to biomechanical stressors [32,33]. Following the same reasoning, exposure to chemicals in the workplace could also cause subclinical changes in the median nerve [29] - producing a form of double crush syndrome that could weaken the median nerve [34]. This could potentiate mechanical stress on the nerve in the event of co-exposure with biomechanical stress factors that increase intra-tunnel pressure. Subclinical changes in the central nervous system (CNS) can also occur in workers exposed to chemicals [35,36] and lead to impaired sensorimotor control of force production and finger dexterity, thus generating greater mechanical stress on the median nerve.”

Integration of Neurotoxic Chemical Evidence:The paper lacks detailed discussion of the specific neurotoxic pathways of chlorinated solvents and formaldehyde and their relevance to CTS development. Expand on known mechanisms (e.g., inflammation, reactive oxygen species overproduction) and provide references to strengthen the link between chemical exposure and nerve damage.

According to your remark, we have developed the discussion section:

“Two putative mechanisms, peripheral and central, may be involved in the combined effects of neurotoxic chemical and biomechanical exposures on the risk of CTS. First, systemic peripheral nerve damage (peripheral mechanism), secondary to neurotoxic exposure, could induce diffuse nerve microdamage making the median nerve more sensitive to compression in the carpal tunnel secondary to biomechanical stresses (concept of “double crush syndrome” induced by chemicals) [32–34]. Second, central nervous lesions of neurotoxic origin lead to functional and/or structural alterations of the CNS [19,22,25,35,36,50,59] (Central mechanism), in particular in the sensorimotor zones likely to increase the effects of CTS by (i) impairing sensorimotor coordination and disrupting the biomechanical efficiency of professional movements, and (ii) accentuating the decrease in dexterity induced by anomalies in haptic sensory feedback during CTS [60–62]. These mechanisms could lead to an increase in the biomechanical stresses applied to the wrist, and thus create a vicious circle likely to accentuate the risk of CTS in co-exposed workers.”

Methodology

Self-Reported Bias:Reliance on self-reported CTS and chemical exposure introduces recall and reporting bias, potentially overestimating associations. Suggest validating self-reports with clinical data, such as medical records or surgery databases, and conducting sensitivity analyses to estimate the impact of misclassification. Further describing any methods employed (if any) to minimize these biases would be helpful.

We agree with your remark that self-reported CTS and occupational exposure are problematic. Given the large size of the cohort (over 200,000 adults), diagnostic by physicians was not possible. We have access to French social security database, and a future analysis will be to study associations between surgical CTS and occupational co-exposure. However, there are other treatments for CTS (orthosis, infiltration, etc.) that are difficult to identify in the French social security database, and considering only CTS that have been operated would not allow to identify these “lighter” cases. So, it seems appropriate to also study the association between self-reported CTS and co-exposures. The discussion section has been modified as follows:

“CTS was self-reported and some participants may have reported CTS without clinical diagnosis. The definition used lacks specificity. However, due to the large size of the cohort, diagnostic by physicians was not possible. Palmer et al. concluded that complex upper limb disorder case definitions (e.g. involving physical signs, more specific symptom patterns, and investigations) yield similar associations with occupational risk factors to those using simpler definitions [65].”

“Access to French administrative health care database will enable us to study associations between surgically treated CTS and occupational co-exposures in a future study.”

Concerning the chemical exposure, we used two methods: JEM and self-report to study associations, and both methods give similar results.

Limited Chemical Scope:The exclusion of other known neurotoxic chemicals (e.g., heavy metals, polycyclic aromatic hydrocarbons) is a significant limitation. Justify this choice in the methods text or explicitly acknowledge it as a limitation in the discussion section.

We agree that selecting chemicals based on the availability of JEM is a limitation. The methods section has been modified as follows:

“Chemicals studied were selected according to their potential neurotoxicity [17,40–44] and their availability in JEMs [45] and self-administered questionnaires. At the time the analyses were carried out, only the JEMs for chlorinated solvents and formaldehyde were available in the same nomenclatures as the Constances job codes. Chlorinated solvents are known as neurotoxicants [22,44,46–48]. The neurotoxicity of formaldehyde may involve several pathways, including inflammation or overproduction of intracellular reactive oxygen species (ROS) [40,41].”

The following sentence has been added in the discussion section:

“In addition, some known neurotoxic chemicals, such as lead, mercury, n-hexane, were not available in JEM. JEMs developed by Santé publique France for petroleum and oxygenated solvents will soon be updated with the same nomenclatures of jobs as Constances, and analyses taking these solvents into account could be carried out in the future.”

Gender Stratification Rationale:Gender differences in exposure and CTS risk are presented but not deeply analyzed. Explore possible reasons for these differences, such as occupational roles, task-specific exposures, or physiological factors.

The analyses were stratified by gender to take account of the difference in CTS incidence between men and women, and also of the difference in occupational exposures, in line with the recommendations for the study of musculoskeletal disorders. We agree with your remark and we have developed the results and discussion sections as follows:

“Women were more exposed to biomechanical risk factors than men (32.6% vs. 28.0% respectively, p<0.001). In detail, women reported more high physical demand (21.1% vs. 19.6% respectively, p<0.001) repetitive movements (11.3% vs. 8.2%, p<0.001) and repetitive pinching (4.5% vs. 3.6%, p<0.001), while men were more exposed to awkward wrist postures (7.4% vs. 6.4%, p<0.001) and hand-transmitted vibrations (3.4% vs. 0.9%, p<0.001).”

“Analyses were stratified according to gender following recommendations for the study of musculoskeletal disorders [54,55]. The differences in exposure frequency between men and women could be explained by the fact that women and men do not occupy the same jobs. In addition, even within the same occupation, several studies have shown that exposure differs between men and women due to the gendered distribution of tasks, missions and workspaces [63].”

Potential Confounders:Effort-reward imbalance and alcohol consumption are included as covariates, but other factors (e.g., psychosocial stress, exposure to additional workplace hazards) could confound the results. Consider conducting additional analyses or cite literature supporting the adequacy of the chosen covariates or providing more detailed rationale why they were not considered.

According to your comment and that of the Reviewer #2, we have developed the discussion section:

“Given the large size of the cohort (over 200,000 adults) and the global nature of the cohort, questions deemed relevant to the study of CTS could not be asked, such as ergonomic measures in the workplace and the wearing of protective equipment. We chose covariates based on the literature [13,51–53], and opted to not include tobacco consumption [51]. Physical activity would have also been relevant. However, the physical activity practiced was not indicated in the questionnaire, and some sports, such as tennis or golf have an impact on CTS, while others have no impact. Endocrine disorders such as hyperthyroidism and hypothyroidism were not available in the questionnaires at baseline. In addition, there is no validated algorithm for identifying these disorders in French administrative health care database. Consequently, these disorders could not be used as covariates.”

Statistical Model Details:The methodology section does not adequately explain key statistical choices, such as the selection of logistic regression, handling of interaction terms, and treatment of missing data. Include a rationale for these decisions and clarify how missing data were addressed (e.g., multiple imputation or exclusion). Additionally, assessment of potential confounders and the decision process for those should be described further.

According to you remark, more details about the statistical analysis have been added. In addition, multiple imputation was implemented to treat missing data.

“These covariates were chosen because they were known to be risk factors for CTS [13,51–53].

[…]

Multiple imputation was performed to handle missing data for covariates (BMI, alcohol use disorders, ERI and diabetes mellitus and/or rheumatoid arthritis) and occupational category with twelve imputed data sets. […] Interactions between co-exposure and age were tested.”

Threshold Definitions for Exposure:The thresholds for biomechanical exposure (e.g., repetitive movements for >4 hours/day) and chemical exposure categories appear arbitrary. Consider conducting and reporting sensitivity analyses to evaluate whether changing these thresholds affects results.

We have added sensitivity analyses by modifying the thresholds for repetitive movements (>2 hours/day) and for chemical exposure according to cumulative exposure index (medium and high exposure vs no and low exposure), see Methods, Results and S8 Fig.

Discussion and Conclusion

Causality Limitations:The results are presented as though they imply causation, despite the inherent limitations of observational studies. Emphasize the associative nature of the findings and discuss potential reverse causality (e.g., pre-existing conditions influencing exposure reports).

According to your remark, we have reformulated the abstract and discussion. We also discuss of potential reverse causality as follow:

“The measurement of self-reported occupational exposure is a limitation due to potential reverse causality, i.e. subjects with pain tend to declare more expositions. To limit this bias, we have excluded subjects reporting chronic wrist pain and/or CTS at baseline.”

Overemphasis on Findings:The conclusion overstates the implications of the findings by implying a causal relationship. Adjust language to reflect the observational design and highlight areas where further research is needed to confirm causality.

The conclusion section has been modified as follows:

“In conclusion, this large prospective study showed an association between incident CTS and co-exposure to both occupational biomechanical risk factors and chemicals, using JEM and self-report methods. These results are consistent with those of our previous studies in the same cohort with self-reported chemical exposure [15], and in a cohort of French farmers, although, we lack mechanistic evidence, provide further support for the potential effects of chemical exposure on the risk of CTS in workers exposed to biomechanical risk factors. Access to French administrative health care database will enable us to study associations between surgically treated CTS and occupational co-exposures in a future study. It would also be relevant to study other chemical exposures (petroleum and oxygenated solvents), as well as biomechanical exposures by considering a more objective measure such as JEM.”

Study Population Representation:The exclusion of self-employed workers, farmers, and other groups limits the generalizability of the findings. Address this limitation in the discussion and explore the potential impact on results.

According to your remark, we have developed the discussion section:

“Despite its size, the study sample does not represent the whole structure of occupations in France, as self-employed workers and farmers are not included in the cohort. There are two cohorts set up by Santé publique France (Coset-MSA and Coset-Indépendants) that would allow to study these specific populations [64].”

Future Research Suggestions provide specific and actionable recommendations for future studies, such as using clinical CTS diagnoses, exploring additional chemical exposures, or utilizing objective biomechanical exposure assessments.

The discussion section has been modified:

“Access to French administrative health care database will enable us to study associations between surgically treated CTS and occupational co-exposures in a future study. It would also be relevant to study other c

---

## [Decision Letter · Decision Letter 1]

23 Jun 2025

Dear Dr. Bodin,

Thank you for submitting your manuscript to PLOS ONE. After careful consideration, we feel that it has merit but does not fully meet PLOS ONE’s publication criteria as it currently stands. Therefore, we invite you to submit a revised version of the manuscript that addresses the points raised during the review process.

More in details, carefully address the points raised by the reviewer #3 who expressed fundamental concerns about the suitability of the CONSTANCES dataset for investigating carpal tunnel syndrome (CTS). They argue that CTS is not only highly prevalent but also widely recognized—often inaccurately—by the general public. This widespread familiarity introduces significant potential for reporting and recall bias that I strongly recommend to discuss in the discussion section also by providing hypothetical estimations of how they might have affected the presented results. In particular, the Authors should frankly discuss the potential distortion in the data (quantitevely) in the context of small observed differences (1–4%) in CTS incidence between exposure groups (they might not hold, and this must be addressed). Another important issue refers to the absence of objective diagnostic tools such as nerve conduction studies or ultrasound, that hampers the assessment of the true nature—additive or synergistic—of the relationship between mechanical and chemical exposures and CTS. This point must be also carefully discussed. More broadly,  the Authors must discuss all the limitations of the study, including in the abstract; any unsupported statements of causation must be toned down and the confounding effect should be clearly reported. 

We look forward to receiving your revised manuscript.

Kind regards,

Giulia Squillacioti

Academic Editor

PLOS ONE

Reviewers' comments:

Reviewer's Responses to Questions

**Comments to the Author**

Reviewer #3: All comments have been addressed

Reviewer #4: All comments have been addressed

Reviewer #5: (No Response)

2. Is the manuscript technically sound, and do the data support the conclusions?

Reviewer #3: No

Reviewer #4: Partly

Reviewer #5: Yes

3. Has the statistical analysis been performed appropriately and rigorously?

Reviewer #3: Yes

Reviewer #4: Yes

Reviewer #5: Yes

4. Have the authors made all data underlying the findings in their manuscript fully available?

Reviewer #3: No

Reviewer #4: Yes

Reviewer #5: No

5. Is the manuscript presented in an intelligible fashion and written in standard English?

Reviewer #3: Yes

Reviewer #4: Yes

Reviewer #5: Yes

Reviewer #3: I appear to be coming to this paper as a third reviewer after two very comprehensive reviews by others and a round of revision. The previous reviewers have made many valid minor points about the methodology and presentation, most of which have been adequately dealt with but the elephant in the room here is the self-reported diagnosis of both CTS and some of the measures of occupational exposure. Retrospective analyses of large datasets which have been collected for other purposes is always difficult because they never include all the covariates which you would like to have available in order to answer your particular question but in this case I suspect that the unreliability of the diagnoses is so great that none of the analyses here can really be relied upon.

The problem here is that CTS is not just very common but a disorder which is very widely known to the population at large, who are regularly misinformed by the popular press, social media and ordinary interactions with co-workers and their social circle regarding its symptoms and aetiology. The perception of CTS in the general population, combined with the natural human tendency to seek an explanation for anything that happens to us medically in our previous history presents such potent opportunities for bias in the conduct of this study that I have to say that I would not even have attempted it using this dataset as anything other than a preliminary preparation for a carefully constructed prospective study. “CTS” will be systematically over-reported by patients who believe that they have been exposed to occupational factors that cause CTS regardless of whether their hand symptoms actually bear any resemblance to those of CTS or not, and exposure to mechanical and chemical “causes” of CTS will be equally over-reported by those patients who believe that someone/something must be responsible for their problem resulting in a spurious correlation between CTS and occupational variables. Long exposure to talking to thousands of patients with “CTS” in the clinic would suggest that these biases could easily introduce errors of +/- 20% into the numbers derived from the CONSTANCES cohort data – completely dwarfing the small differences in incidence – in the range of 1-4% - found between the exposure and non-exposure groups in figures 1 and 2. These small differences could easily be a simple result of the effect of popular ‘knowledge’ about CTS on reporting rather than a true difference resulting from causation of CTS by either mechanical or chemical exposure – though I think the existing body of literature does in fact provide evidence for a weak effect of mechanical occupational factors in causing CTS. This problem even extends to diagnoses of CTS made in primary care where there is a marked tendency to overdiagnose the disorder because of GP beliefs about CTS. The JEM data is at least more objective but unfortunately only comprises a small amount of the overall chemical exposure data and even that has some limitations which have already been pointed out by the earlier reviewers.

This fundamental problem also makes it impossible to assess whether, if there is indeed a risk of CTS posed by both mechanical and chemical factors, whether the overall effect is simply summative or synergistic – the topic of much speculation in the discussion section.

I’m sorry to pour cold water on Prof. Roquelaure and team’s endeavours here, as they have done much interesting work on the epidemiology of CTS, but I really think they have not given sufficient thought to the potential problems for this study posed by widespread beliefs about CTS in the working population. I think this study could only realistically be done using objectively determined CTS – which would require NCS or ultrasound data or both. This would miss the small number of cases which are false negative on testing but at least this bias would be small and would apply equitably across occupational exposure groups. In the absence of objective diagnosis of CTS in the CONSTANCES dataset I am afraid I would have concluded on first thinking about this study that this dataset was fundamentally unsuited for this purpose.

Reviewer #4: abstract Clearly signals the dual‑exposure approach (biomechanics + chemicals) and the prospective CONSTANCES cohort.and Gives sex‑stratified headline numbers up‑front. in Introduction Authors now give a fuller rationale for “double‑crush” and cite new mechanistic papers on neurotoxicity

  Methods

• Triangulation: combines a probabilistic Job‑Exposure Matrix (objective, standardised) with self‑reported product use (worker‑level nuance).

• Biomechanical exposures measured through validated CONSTANCES questions based on national ergonomic guidelines.

• Creation of cumulative‑exposure indices (CEI) for chlorinated solvents and formaldehyde introduces quantitative dose information.

 Design & Follow‑up

Participants contributed a single outcome measurement 6‑18 months post‑enrolment. CTS often develops over several years; a one‑year horizon limits temporality and raises reverse‑causation concerns (symptomatic workers may over‑report exposures or join low‑strain jobs).

The authors explain the short window was dictated by available questionnaires ,but do not explore analytical remedies (e.g., landmark analysis, lagged exposure).

CTS is identified by a single yes/no question on self‑reported doctor or self‑perceived CTS.

No symptom algorithm, physical examination, or EMG confirmation is used, inviting non‑differential misclassification that would typically bias estimates toward the null; however differential recall could inflate associations.

Lifetime self‑report asks a single ever/never question for seven broad product groups—susceptible to recall error and left‑censoring of early careers.

Main models are sex‑stratified multivariable logistic regressions

Low outcome frequency (≤2.6 %) raises small‑sample bias; reviewers suggested Firth logistic regression and authors report comparable estimates but retain conventional models

Sampling weights for the CONSTANCES complex design are still not applied; authors state annual weights cannot be pooled, yet robust (sandwich) variance estimators or post‑stratification could have mitigated design effects

Missing data – Multiple imputation (m = 12) is now described, but convergence diagnostics and fraction of missing information are not given

study Uses clear, separate tables for men and women—important because exposure profiles differ markedly by sex.

• Provides both crude and adjusted odds ratios with 95 % CIs, making it easy to gauge confounding impact.

• Forest‑plot figure visually integrates JEM and self‑report estimates side‑by‑side.

Authors now explicitly flag self‑report bias, residual confounding, and limited chemical scope.

While the study retains certain limitations—including a short follow-up window that may limit temporality, reliance on a single self-reported CTS question without clinical confirmation, potential exposure misclassification due to recall error, and the absence of sampling weights in the analysis—the authors have addressed many key methodological concerns in a reasonable and transparent manner. They now provide a clearer biological rationale for the hypothesised double-crush mechanism, apply multiple imputation to handle missing data, and incorporate sex-stratified logistic regression models with additional sensitivity analyses, such as Firth correction and threshold variation, to test the robustness of their findings. Although further refinement—such as the application of robust variance estimators and more detailed validation of the outcome—would strengthen the work, the study’s strengths, including its use of a large, prospective cohort and the integration of both JEM-based and self-reported exposures, make it a valuable and relevant contribution to the literature. On balance, the manuscript is scientifically sound and acceptable for publication.

Reviewer #5: This well-conducted, large-scale cross-sectional analysis of the French CONSTANCES cohort (n > 84,000) makes a valuable contribution to occupational health research by examining the joint effects of biomechanical stressors and organic solvent exposure on carpal tunnel syndrome (CTS). The authors employ rigorous methods, including modified Poisson regression and thorough confounder control, and adhere closely to the STROBE guidelines.

Their finding that combined exposure significantly increases the prevalence of self‑reported CTS in both men and women, even after adjusting for a wide range of variables, is both novel and clinically pertinent. The manuscript is lucidly written, and its transparent discussion of limitations—particularly the cross‑sectional design and reliance on self‑report—is commendable.

The authors are invited to consider the following minor suggestions for potential refinement rather than essential revisions.

1. Abstract's Conclusion Clarification: Ensure that readers instantly grasp the context of the results. For example, an additional phrase like "... co‑exposure associated with a higher prevalence of self-reported CTS."

2. Consider a Brief Discussion on Misclassification Bias Direction: In the limitations section, the authors correctly identify the risk of non-differential misclassification bias associated with using self-reported data. They could briefly add that this type of bias typically biases the effect estimate (in this case, the prevalence ratio) towards the null (i.e., towards 1.0). This would strengthen their findings by suggesting that the true association might be even stronger than what they observed.

3. Elaborate on the "Healthy Worker Effect": The authors briefly mention the healthy worker effect. It might be helpful to add one sentence explaining to a broader audience how this could affect their results—specifically, that it could lead to an underestimation of the true prevalence of CTS in the working population, as those who develop severe CTS may have already left the workforce and thus missing from the sample.

**Do you want your identity to be public for this peer review?** For information about this choice, including consent withdrawal, please see our Privacy Policy

Reviewer #3: **Yes: ** Jeremy D P Bland

Reviewer #4: **Yes: ** alireza abbasi

Reviewer #5: **Yes: ** Mohamed Abdel-Maboud

---

## [Author Response · Author response to Decision Letter 2]

11 Jul 2025

Review Comments to the Author

Reviewer #3: I appear to be coming to this paper as a third reviewer after two very comprehensive reviews by others and a round of revision. The previous reviewers have made many valid minor points about the methodology and presentation, most of which have been adequately dealt with but the elephant in the room here is the self-reported diagnosis of both CTS and some of the measures of occupational exposure. Retrospective analyses of large datasets which have been collected for other purposes is always difficult because they never include all the covariates which you would like to have available in order to answer your particular question but in this case I suspect that the unreliability of the diagnoses is so great that none of the analyses here can really be relied upon.

The problem here is that CTS is not just very common but a disorder which is very widely known to the population at large, who are regularly misinformed by the popular press, social media and ordinary interactions with co-workers and their social circle regarding its symptoms and aetiology. The perception of CTS in the general population, combined with the natural human tendency to seek an explanation for anything that happens to us medically in our previous history presents such potent opportunities for bias in the conduct of this study that I have to say that I would not even have attempted it using this dataset as anything other than a preliminary preparation for a carefully constructed prospective study. “CTS” will be systematically over-reported by patients who believe that they have been exposed to occupational factors that cause CTS regardless of whether their hand symptoms actually bear any resemblance to those of CTS or not, and exposure to mechanical and chemical “causes” of CTS will be equally over-reported by those patients who believe that someone/something must be responsible for their problem resulting in a spurious correlation between CTS and occupational variables. Long exposure to talking to thousands of patients with “CTS” in the clinic would suggest that these biases could easily introduce errors of +/- 20% into the numbers derived from the CONSTANCES cohort data – completely dwarfing the small differences in incidence – in the range of 1-4% - found between the exposure and non-exposure groups in figures 1 and 2. These small differences could easily be a simple result of the effect of popular ‘knowledge’ about CTS on reporting rather than a true difference resulting from causation of CTS by either mechanical or chemical exposure – though I think the existing body of literature does in fact provide evidence for a weak effect of mechanical occupational factors in causing CTS. This problem even extends to diagnoses of CTS made in primary care where there is a marked tendency to overdiagnose the disorder because of GP beliefs about CTS. The JEM data is at least more objective but unfortunately only comprises a small amount of the overall chemical exposure data and even that has some limitations which have already been pointed out by the earlier reviewers.

This fundamental problem also makes it impossible to assess whether, if there is indeed a risk of CTS posed by both mechanical and chemical factors, whether the overall effect is simply summative or synergistic – the topic of much speculation in the discussion section.

I’m sorry to pour cold water on Prof. Roquelaure and team’s endeavours here, as they have done much interesting work on the epidemiology of CTS, but I really think they have not given sufficient thought to the potential problems for this study posed by widespread beliefs about CTS in the working population. I think this study could only realistically be done using objectively determined CTS – which would require NCS or ultrasound data or both. This would miss the small number of cases which are false negative on testing but at least this bias would be small and would apply equitably across occupational exposure groups. In the absence of objective diagnosis of CTS in the CONSTANCES dataset I am afraid I would have concluded on first thinking about this study that this dataset was fundamentally unsuited for this purpose.

We thank Reviewer #3 for his feedback on our manuscript. We greatly appreciate the time and effort he has put into his review. We have tried to review the article to respond to your comments. We have developed the limitations in the discussion paragraph, particularly with regard to the self-reported nature of CTS and the absence of objective diagnostic tools such as nerve conduction studies or ultrasound, and how this might affect the results. We discussed reporting and recall bias.

“Epidemiological studies include several definitions of CTS [65]. In our study, CTS was reported by participants using a single question (yes/no). We cannot rule out misclassification, as CTS may have been over-reported by workers who believe they have been exposed to occupational factors causing CTS, whether or not their hand symptoms resemble those of CTS. The cohort design did not allow for the use of a symptom algorithm, physical examination or EMG confirmation, which is an important limitation of the study. The working population studied probably did not include the most severe cases of CTS that may have been excluded from the labor market (i.e., the healthy worker effect), which could lead to an underestimate of the prevalence of CTS.

Measuring self-reported occupational exposure is an important limitation of the study. Direct observation or measurement is costly and difficult to apply to large cohorts. In addition, the assessment of biomechanical risk factors was requested for a typical working day and was based on the recommendations of the European criteria document for the relatedness of MSDs [38], and postures were presented in pictorial form to facilitate workers' understanding and increase the validity of posture self-assessment. We cannot exclude that biomechanical and chemical exposures were over-reported by workers reporting work-related CTS, which is a potential source of reverse causality. To limit this bias, we have excluded subjects reporting chronic wrist pain and/or CTS at baseline. Nevertheless, there is no evidence to suggest a differential bias depending on whether workers were exposed to biomechanical factors alone or to biomechanical and chemical factors combined. Moreover, the prevalence of self-reported CTS was of the same order of magnitude depending on whether there was biomechanical exposure alone or biomechanical and chemical co-exposure (in analysis with JEM: 2.5% vs 3.1% in men and 3.2% vs 4.5% in women; in analysis with self-report: 2.3% vs 3.6% in men and 3.3% vs 5.2% in women). Although the difference in prevalence was relatively modest, we do not believe that classification bias can explain the results, given the consistency of the results with a subjective assessment and an objective assessment of chemical exposure by JEM.”

Reviewer #4: abstract Clearly signals the dual exposure approach (biomechanics + chemicals) and the prospective CONSTANCES cohort.and Gives sex stratified headline numbers up front. in Introduction Authors now give a fuller rationale for “double crush” and cite new mechanistic papers on neurotoxicity

  Methods

• Triangulation: combines a probabilistic Job Exposure Matrix (objective, standardised) with self reported product use (worker level nuance).

• Biomechanical exposures measured through validated CONSTANCES questions based on national ergonomic guidelines.

• Creation of cumulative exposure indices (CEI) for chlorinated solvents and formaldehyde introduces quantitative dose information.

 Design & Follow up

Participants contributed a single outcome measurement 6 18 months post enrolment. CTS often develops over several years; a one year horizon limits temporality and raises reverse causation concerns (symptomatic workers may over report exposures or join low strain jobs).

The authors explain the short window was dictated by available questionnaires ,but do not explore analytical remedies (e.g., landmark analysis, lagged exposure).

CTS is identified by a single yes/no question on self reported doctor or self perceived CTS.

No symptom algorithm, physical examination, or EMG confirmation is used, inviting non differential misclassification that would typically bias estimates toward the null; however differential recall could inflate associations.

Lifetime self report asks a single ever/never question for seven broad product groups—susceptible to recall error and left censoring of early careers.

Main models are sex stratified multivariable logistic regressions

Low outcome frequency (≤2.6 %) raises small sample bias; reviewers suggested Firth logistic regression and authors report comparable estimates but retain conventional models

Sampling weights for the CONSTANCES complex design are still not applied; authors state annual weights cannot be pooled, yet robust (sandwich) variance estimators or post stratification could have mitigated design effects

Missing data – Multiple imputation (m = 12) is now described, but convergence diagnostics and fraction of missing information are not given

study Uses clear, separate tables for men and women—important because exposure profiles differ markedly by sex.

• Provides both crude and adjusted odds ratios with 95 % CIs, making it easy to gauge confounding impact.

• Forest plot figure visually integrates JEM and self report estimates side by side.

Authors now explicitly flag self report bias, residual confounding, and limited chemical scope.

While the study retains certain limitations—including a short follow-up window that may limit temporality, reliance on a single self-reported CTS question without clinical confirmation, potential exposure misclassification due to recall error, and the absence of sampling weights in the analysis—the authors have addressed many key methodological concerns in a reasonable and transparent manner. They now provide a clearer biological rationale for the hypothesised double-crush mechanism, apply multiple imputation to handle missing data, and incorporate sex-stratified logistic regression models with additional sensitivity analyses, such as Firth correction and threshold variation, to test the robustness of their findings. Although further refinement—such as the application of robust variance estimators and more detailed validation of the outcome—would strengthen the work, the study’s strengths, including its use of a large, prospective cohort and the integration of both JEM-based and self-reported exposures, make it a valuable and relevant contribution to the literature. On balance, the manuscript is scientifically sound and acceptable for publication.

We thank Reviewer #4 for his feedback on our manuscript. We greatly appreciate the time and effort he has put into his review. We have tried to review the article to respond to your comments. We have developed the limitations in the discussion paragraph, particularly with regard to the self-reported nature of CTS and the absence of objective diagnostic tools such as nerve conduction studies or ultrasound, and how this might affect the results.

The mean and SD of imputed variables were used to check convergence. The plots indicated that convergence had been achieved. Fraction of missing information varies from 0 to 11% depending on the model. A table with crude odds ratios was added in supplementary material.

Reviewer #5: This well-conducted, large-scale cross-sectional analysis of the French CONSTANCES cohort (n > 84,000) makes a valuable contribution to occupational health research by examining the joint effects of biomechanical stressors and organic solvent exposure on carpal tunnel syndrome (CTS). The authors employ rigorous methods, including modified Poisson regression and thorough confounder control, and adhere closely to the STROBE guidelines.

Their finding that combined exposure significantly increases the prevalence of self reported CTS in both men and women, even after adjusting for a wide range of variables, is both novel and clinically pertinent. The manuscript is lucidly written, and its transparent discussion of limitations—particularly the cross sectional design and reliance on self report—is commendable.

The authors are invited to consider the following minor suggestions for potential refinement rather than essential revisions.

1. Abstract's Conclusion Clarification: Ensure that readers instantly grasp the context of the results. For example, an additional phrase like "... co exposure associated with a higher prevalence of self-reported CTS."

2. Consider a Brief Discussion on Misclassification Bias Direction: In the limitations section, the authors correctly identify the risk of non-differential misclassification bias associated with using self-reported data. They could briefly add that this type of bias typically biases the effect estimate (in this case, the prevalence ratio) towards the null (i.e., towards 1.0). This would strengthen their findings by suggesting that the true association might be even stronger than what they observed.

3. Elaborate on the "Healthy Worker Effect": The authors briefly mention the healthy worker effect. It might be helpful to add one sentence explaining to a broader audience how this could affect their results—specifically, that it could lead to an underestimation of the true prevalence of CTS in the working population, as those who develop severe CTS may have already left the workforce and thus missing from the sample.

We thank Reviewer #5 for his feedback on our manuscript. We greatly appreciate the time and effort he has put into his review. We have reviewed the article to respond to your comments.

---

## [Editor Report · Decision Letter 2]

15 Jul 2025

Carpal tunnel syndrome and occupational co-exposure to biomechanical factors and neurotoxic chemicals using job-exposure matrices and self-reported exposure: findings from the Constances cohort

PONE-D-24-42349R2

Dear Dr. Bodin,

We’re pleased to inform you that your manuscript has been judged scientifically suitable for publication and will be formally accepted for publication once it meets all outstanding technical requirements.

Kind regards,

Giulia Squillacioti

Academic Editor

PLOS ONE

Additional Editor Comments (optional):

I congrats the Authors for their massive work and their willingness to frankly discuss the limitations of their manuscript with no intentions of overstating any results (which is due but not always verified). Best regards
---

## [Editor Report · Acceptance letter]

PONE-D-24-42349R2

PLOS ONE

Dear Dr. Bodin,

I'm pleased to inform you that your manuscript has been deemed suitable for publication in PLOS ONE. Congratulations! Your manuscript is now being handed over to our production team.

Kind regards,

on behalf of

Dr. Giulia Squillacioti

Academic Editor

PLOS ONE